# Musical expertise generalizes to superior temporal scaling in a Morse code tapping task

**Matthew A. Slayton** [id]1‡, **Juan L. Romero-Sosa**2,3‡, **Katrina Shore**1, **Dean V. Buonomano**2,3,4‡ *, **Indre V. Viskontas**1,5‡ *

**1** San Francisco Conservatory of Music, San Francisco, CA, United States of America, **2** Department of Neurobiology, University of California Los Angeles, Los Angeles, CA, United States of America, **3** Neuroscience Interdepartmental Program, University of California Los Angeles, Los Angeles, CA, United States of America, **4** Department of Psychology, University of California Los Angeles, Los Angeles, CA, United States of America, **5** Department of Psychology, University of San Francisco, San Francisco, CA, United States of America

‡ MAS and JRS are Joint First Authors on this work. DVB and IVV are Joint Senior Authors on this work.
* dbuono@ucla.edu (DVB); ivviskontas@usfca.edu (IVV)

**Data Availability Statement:** The data underlying the results presented in the study are freely available on Open Science Framework at https://osf.io/hf6k5/.

## Abstract

A key feature of the brain's ability to tell time and generate complex temporal patterns is its capacity to produce similar temporal patterns at different speeds. For example, humans can tie a shoe, type, or play an instrument at different speeds or tempi—a phenomenon referred to as temporal scaling. While it is well established that training improves timing precision and accuracy, it is not known whether expertise improves temporal scaling, and if so, whether it generalizes across skill domains. We quantified temporal scaling and timing precision in musicians and non-musicians as they learned to tap a Morse code sequence. We found that non-musicians improved significantly over the course of days of training at the standard speed. In contrast, musicians exhibited a high level of temporal precision on the first day, which did not improve significantly with training. Although there was no significant difference in performance at the end of training at the standard speed, musicians were significantly better at temporal scaling—i.e., at reproducing the learned Morse code pattern at faster and slower speeds. Interestingly, both musicians and non-musicians exhibited a Weber-speed effect, where temporal precision at the same absolute time was higher when producing patterns at the faster speed. These results are the first to establish that the ability to generate the same motor patterns at different speeds improves with extensive training and generalizes to non-musical domains.

## Introduction

Central to the brain's ability to perform precise sensory and motor tasks is the capacity to execute these tasks at varying speeds [1–7]. This ability to temporally scale motor responses is critical and intrinsic to many motor behaviors, such as catching a ball, speaking, or playing a musical instrument. While the neural mechanisms underlying timing on the scale of hundreds of milliseconds to seconds continue to be debated, there is converging evidence from animal

**Funding:** The authors received no specific funding for this work.

**Competing interests:** The authors have declared that no competing interests exist.

studies suggesting that some forms of timing are encoded in dynamically changing patterns of neural activity, or population clocks [4,8–13]. Importantly, experimental evidence and simulations suggest that these patterns of activity can unfold at different speeds, thus potentially accounting for the temporal scaling of motor responses [4,6,14,15].

A well-established characteristic of motor timing is that it operates in accordance with Weber's Law, whereby the variability of a timed response is proportional to the length of the interval [16–18]. It has also been shown that temporal precision can be improved by subdividing a desired interval [19–21] or by increasing movement speed [6]. For example, the variability of a response produced at one second is smaller if it is embedded within a fast rather than a slow pattern—a phenomenon called the *Weber-speed effect* [6].

Numerous studies have established that the relationship between temporal variability (as measured by standard deviation) of a response and the mean time of the response—i.e., the Weber Coefficient—improves with practice [22–25]. Consistent with these observations, musicians generally exhibit superior performance in a number of temporal, sensory, and motor tasks [26–30]. What has not been investigated, however, is whether temporal scaling improves with practice. For example, once a temporal motor pattern is learned, is it the case that it can be produced accurately at different speeds, similar to increasing or decreasing the playback speed of a movie? Or, is the ability to produce a given pattern at different speeds itself learned and therefore dependent on experience?

Here, we investigated whether musicians, who are trained specifically to have timing expertise, exhibit superior temporal scaling on a nonmusical temporal pattern reproduction task [6]. Specifically, we used a Morse code production task because it is purely temporal and non-rhythmic. Additionally, we explored whether the Weber-speed effect is still present in musicians, or rather, as might be desirable for experts, whether temporal precision is constant across different speeds.

## Results

We trained a group of highly experienced musicians and a group of non-musician controls on a temporal pattern reproduction task in which subjects listened to the word "time" in Morse code and attempted to reproduce a sequence of six dots and dashes as accurately as possible (Fig 1A). After each trial, they received visual feedback about the difference between the produced and target patterns, as well as a score representing the correlation between the two. Both groups repeated this task over four successive days, and on the fifth day subjects were tested on their ability to reproduce the pattern at three different speeds: the original speed (1x), as well as twice (2x) and half (0.5x) the trained speed.

To quantify learning we determined the Weber Coefficient across training days by plotting the standard deviation of the tap times against the mean tap times and calculating the slope of the linear regression (Fig 1B, single subject). As measured by the normalized root-mean-square-error (NRMSE), performance improved in both groups (day effect of a two-way ANOVA, $F_{3,48} = 11.4$, $p < .001$; no significant group or interaction effect) (Fig 1C). This improvement reflects, in part, learning of the Morse code pattern itself. Interestingly, controls showed a significant decrease in the Weber Coefficient over the four days of training ($F_{1,3} = 7.65$, $p < .001$), whereas musicians started with a low Weber Coefficient that did not improve over time ($F_{1,3} = .09$, $p = .967$) (Fig 1D). These results establish that while both groups learned the pattern over time, the musicians exhibited a floor effect for temporal precision—i.e., consistent with previous studies, their cross-trial variance was low from the outset.

Next, we asked whether there were any differences in overall performance or temporal precision between both groups on the 1x pattern on test day (Day 5). In this phase of the study,

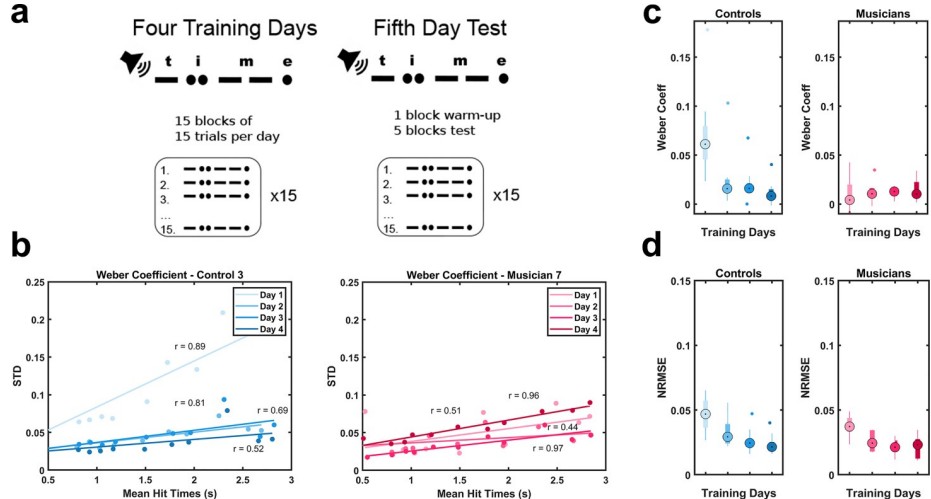

**Fig 1. Temporal precision in non-musician controls but not musicians improves significantly over the course of four training days.** (A) Schematic of the protocol and stimuli. (B) Sample data across four days for one musician and one control subject. The slope of the linear fit of the standard deviation versus mean tap time was defined as the Weber Coefficient. (C, D) Boxplots were plotted for all nine subjects from each group for the four training days for both normalized root-mean-square-error (NRMSE) (C) and Weber Coefficient (D).

subjects were asked to produce the pattern under freeform conditions—i.e., in blocks in which they did not listen to the pattern first. To quantify the performance on test day we calculated the Weber Coefficient of the averaged tap times (Fig 2A) and NRMSE (Fig 2B). There was no significant difference between musicians and controls in their ability to produce the learned pattern at 1x speed, suggesting that the control group was able to reach a similar level of mastery in the task before we tested the temporal scaling capabilities of both groups.

## Musicians exhibit better temporal scaling

We next quantified temporal scaling–how well subjects produced the pattern at double (2x) and half speed (0.5x) on test day. Because the responses generally differed significantly from the targeted 2x and 0.5x speeds, we first quantified the actual magnitude of the speed-up or slow-down. To this end, we estimated a Scaling Factor based on the ratio of the 1x duration to the 2x and 0.5x durations. Perfect scaling would be 2 and 0.5 for the fast and slow speed conditions, respectively. There was no significant difference in the Scaling Factor between groups ($F_{1,16} = .203$, $p = .66$) (Fig 3A). Both musicians and controls produced the pattern at approximately the same speeds: 1.6x and 0.7x in the 2x and 0.5x conditions, respectively.

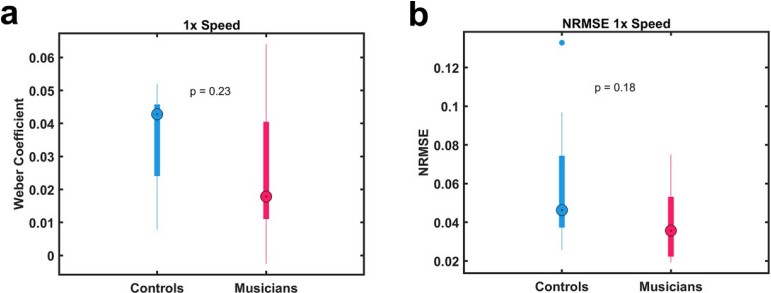

**Fig 2. Performance at the 1x speed on test day.** (A, B) On test day there was no significant difference between the musician and control groups for either the Weber Coefficient (A) or the NRMSE (B) at the 1x speed.

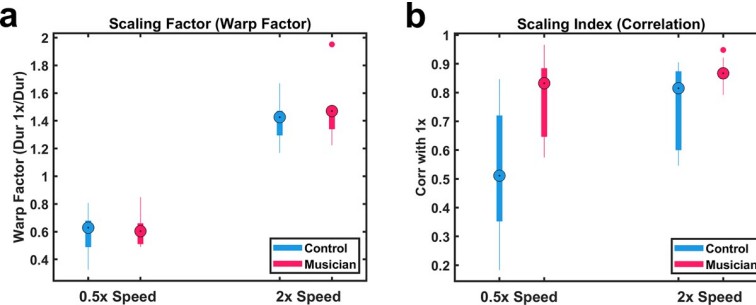

**Fig 3. Superior temporal scaling in musicians.** (A) Magnitude of temporal scaling as measured by Scaling Factor, the ratio of the duration of the 1x pattern in relation to the duration of the 0.5x or 2x patterns. (B) Quality of temporal scaling as measured by Scaling Index (the correlation between the mean tap times in the 2x or 0.5x and 1x patterns). Scaling Index was significantly better in the musician group at both speeds.

Having established the absence of any differences in the speeds produced, we next quantified the quality of the temporal scaling. We defined a Scaling Index as the correlation between the mean tap times at the 2x or 0.5x speeds with the taps produced at the 1x speed—thus perfect scaling at any speed would correspond to a Scaling Index of 1. A two-way ANOVA with repeated measures on one factor revealed a significant difference between speeds ($F_{1,16} = 8.57$, $p = .009$) (Fig 3B), i.e., the correlation between the 2x taps and 1x taps was significantly higher than the correlation between the 0.5x taps and 1x taps. Together with the lack of an interaction effect ($F_{1,16} = 1.91$, $p = .19$), these findings indicate that both groups were better at scaling to faster speeds. Importantly, there was also a significant group effect ($F_{1,16} = 11.93$, $p = .003$). Thus, although the overall performance at the 1x condition was similar and there was no difference in the produced speeds, the musicians were significantly better at scaling the motor pattern to both faster and slower speeds. Importantly, the difference in the quality of temporal scaling cannot be attributed to the magnitude of the scaling (i.e., how fast or slow the patterns were) because there was no group difference in the speeds of the patterns in either the 2x or 0.5x condition.

### The Weber-speed effect was observed in both groups

To determine whether a Weber-speed effect were present, and if the effects were similar or different in musicians and controls, we estimated the Weber Coefficient (Fig 4). A two-way ANOVA with repeated measures on one factor revealed a significant difference in speed ($F_{1,16} = 12.585$, $p = .0027$) and the absence of an interaction. Thus, subjects in both groups were more precise at the 2x speed—demonstrating the Weber-speed effect in both groups. There was also a significant difference between groups ($F_{1,16} = 5.273$, $p = .0355$), but no interaction. Finally, as shown in Fig 4C, the coefficient of variation (the per standard deviation mean tap time ratio) was significantly lower across all taps in the musician group for both the 2x ($F_{1,160} = 9.8$, $p < .001$) and the 0.5x ($F_{1,160} = 9.8$, $p < .001$). These findings provide additional support for those in Fig 3, and further establish that the temporal precision at the scaled speeds was superior in the musician group.

### Discussion

The ability to execute well timed movements at different speeds is a fundamental feature of motor control. Our results provide the first evidence that the ability to speed up or slow down motor patterns precisely is dependent on experience and generalizes across skill domains. Specifically, even though the temporal precision of motor patterns was equivalent in both the

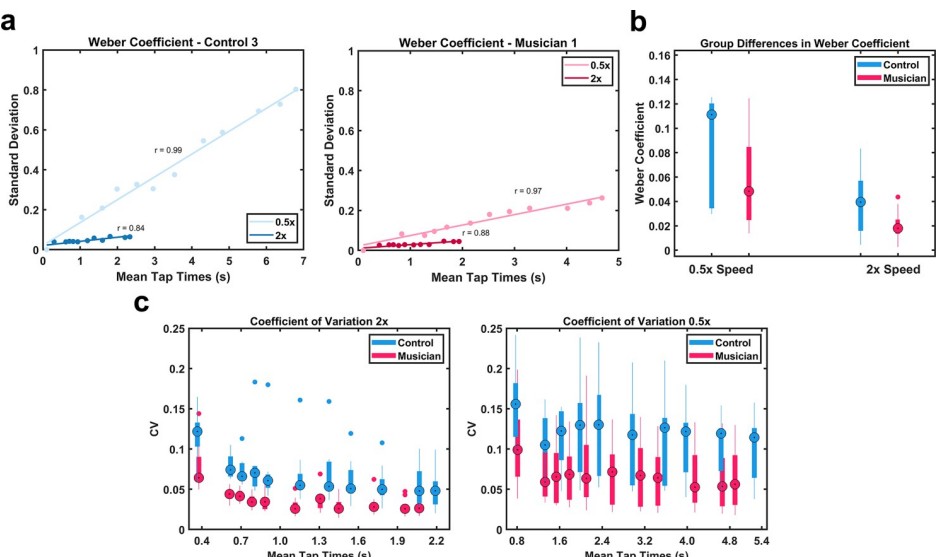

**Fig 4. Analysis of variance at scaled times.** (A) Example of the Weber Coefficient for a musician and a control subject at the 0.5x and 2x speeds. (B) Analysis of the Weber coefficients revealed main effects of group (Musicians x Controls) and speed (0.5x versus 2x). (C) The coefficient of variation (the ratio of standard deviation and mean) across all taps was significantly lower for the musician group at both speeds.

musician and non-musician groups at the end of training (Fig 2), the temporal precision of the musicians on a nonrhythmic temporal pattern production task was significantly better when they were asked to produce the trained pattern at faster and slower speeds (Fig 3).

Additionally, our results are consistent with previous studies demonstrating that, at baseline, musicians generally exhibit superior temporal precision and accuracy in both the sensory and motor domains [21,29–34]. For example, on the first day of training, the Weber Coefficient of the musicians was dramatically smaller (Fig 1). The neural mechanisms underlying improved temporal precision in musicians are not known, but they likely include higher-order cognitive strategies as well as unconscious motor skills. For example, musical training has been shown to promote endogenous oscillations, which may endow musicians with superior ability to track the beat and predict the timing of beat onsets [35]. Training also facilitates greater synchronization with partners across different spontaneous timing rates, with musicians showing lower variability than non-musicians due to greater engagement of error-correction strategies [36,37]. It has also been observed that covariance between fingers was highly constant across tempi in expert pianists [30]. Finally, the earlier the age at which musicians start training, the better they perform on tests of timing expertise [38].

The cognitive mechanisms underlying temporal scaling are even less understood than those underlying timing in general. However, one possible mechanism by which humans may accomplish scaling is through the synergistic recruitment of sensory and motor areas [39,40]. Auditory-motor integration, for example, is associated with the capacity to identify the temporal structure of rhythm [34]. Rhythm training also strengthens inhibitory control, potentially eliminating unnecessary actions by fine-tuning motor networks that track the temporal structure of music [24]. Furthermore, it should be stressed that the relative increase in accuracy with increased speed is distinct from the standard speed (reaction time)-accuracy trade-off, a separate phenomenon that is generally studied in reaction time tasks [41,42] in which performance decreases with speed.

At a more mechanistic level, it has been proposed that timing of complex motor patterns relies on population clocks—i.e., time-varying changes in the spatiotemporal patterns of

activity that can encode time and drive motor activity [10,43]. An attractive property of the population clock model is that it can account for temporal scaling if the speed at which these patterns flow can be modulated. There is indeed evidence that, in the case of the production of simple intervals or durations, temporally scaled patterns of activity are observed as animals produce intervals of different durations [4,11,14,15]. Furthermore, there is theoretical evidence that recurrent neural networks can be trained to account for temporal scaling of complex motor patterns by changing the speed of neural patterns of activity [6].

A somewhat counterintuitive property of motor timing is that temporal precision can be improved by subdivision [20,21,44]—e.g., to produce a more precise 1 second duration, humans can subdivide the response into two intervals of 0.5 seconds. Relatedly, temporal precision is also improved by increasing motor speed [6]. This Weber-speed effect is somewhat paradoxical in that it implies that temporal precision is best when generating a temporal pattern at a fast speed, or while playing a musical piece at a fast tempo (i.e., the Weber Coefficients are smaller at fast speeds). In other words, both absolute and relative timing are significantly worse when playing a musical piece at a slow tempo. Here we asked whether training might allow musicians to overcome the Weber-speed effect, as it might be desirable to equate and maximize temporal precision at all speeds. We determined that this is not the case. In both musicians and non-musicians, the Weber Coefficients were significantly smaller in the 2x condition, and the ratios between the 2x and 0.5x conditions were similar as indicated by the absence of a significant interaction (Fig 4). These results suggest that the Weber-speed effect is an intrinsic and perhaps unavoidable consequence of the neural mechanisms underlying timing, a hypothesis that is consistent with the observation that computational models of timing that rely on the dynamics of recurrent neural networks also exhibit the Weber-speed effect [6].

## Materials and methods

### Recruitment of subjects

Musicians and non-musicians were recruited to participate in the study. The musician group comprised ten subjects, one of whom was excluded because of a data collection error. The musicians included seven graduate students at the San Francisco Conservatory of Music, two professional musicians who had previously earned a Master of Music degree, and one UCLA undergraduate majoring in music. They had each been playing music for an average of 14.2 years (SD = 2.97 years) and their ages ranged from 19 to 28 years old. The non-musician group included ten undergraduate students from University of San Francisco and UCLA whose ages ranged from 18 to 24 years old. Although six non-musician subjects had played music in the past, none were studying music at a university level nor actively practicing or performing. All experiments were run in accordance with the University of California Human Subjects Guidelines and were approved by UCLA's Institutional Review Board. Participants provided written informed consent before participation and were paid $10/hour for their participation.

### Protocol description

Subjects participated in the experiment for five consecutive days. During training days, subjects heard and reproduced the same pattern by tapping a keypad in 15 blocks of 15 repetitions each (Fig 1A). After each trial, they saw a visual representation of the timing of the produced and target patterns, as well as a score representing the correlation. On test day (Day 5), subjects were given one block as a warm-up, and then were asked to produce the pattern at three speeds (in randomized order): twice as fast (2x), half as fast (0.5x), and at the original speed (1x) without receiving any additional auditory stimuli. The target pattern was a series of short and long 550Hz tones, spelling the word "time" in Morse Code. When prompted by written instructions

on the screen, subjects played back the pattern using a single button on a Cedrus Response Pad™, connected to a personal computer using custom Matlab code and the Psychophysics Toolbox. The target speed for the Morse Code was 10 WPM (words per minute). Every 'dot' was 120ms long and every 'dash' was 360ms long. Between each element that comprises a letter was a 'dot-length' pause, and between the last element of a letter and the first element of another letter was a 'dash-length' pause for a total duration of 2.76s. The specific sequence was dash (0-360ms) + pause (360-720ms) + dot (720-840ms) + pause (840-960ms) + dot (960-1080ms) + pause (1080-1440ms) + dash (1440-1800ms) + pause (1800-1920ms) + dash (1920-2280ms) + pause (2280-2640ms) + dot (2640-2760ms). After the reproduction, a visual representation of the target stimulus as well as the subject's entry appeared on the screen, alongside a score reflecting the correlation between the target and reproduced pattern. Between blocks, participants were allowed to take a short break. However, during each block, participants were asked to interact with the program continuously. The stimulus generation and experimental interface were written in Matlab. Data analysis and statistics relied on custom written Matlab code.

### Weber Coefficient and data analysis

The Weber Coefficient was calculated as described previously using Weber's generalized Law [45,46], in which the relationship of mean and variance of motor responses is represented as $\sigma^2 = kT^2 + \sigma^2_{independent}$ where $\sigma^2$ represents total variance, $\sigma^2_{independent}$ the tempo-independent variance, $k$ the Weber Coefficient that approximates the square root of the conventional Weber fraction at long intervals, and $T$ is time. Trials were excluded according to two sequential criteria: when the end time of the produced pattern was three or more times longer than the target end time (generally indicating that a subject lost track of how many taps they produced), and next when the values were more than two standard deviations from their mean. The Normalized Root Mean Squared Error was calculated by finding the square root of the means of the squared differences between produced time and target time, normalized by the time of the final target time tap offset.

## Author Contributions

**Conceptualization:** Juan L. Romero-Sosa, Katrina Shore, Dean V. Buonomano.

**Data curation:** Matthew A. Slayton, Juan L. Romero-Sosa.

**Formal analysis:** Matthew A. Slayton, Juan L. Romero-Sosa.

**Investigation:** Katrina Shore.

**Methodology:** Juan L. Romero-Sosa, Katrina Shore.

**Project administration:** Indre V. Viskontas.

**Software:** Matthew A. Slayton, Juan L. Romero-Sosa.

**Supervision:** Indre V. Viskontas.

**Writing – original draft:** Matthew A. Slayton, Dean V. Buonomano, Indre V. Viskontas.

**Writing – review & editing:** Matthew A. Slayton, Dean V. Buonomano, Indre V. Viskontas.

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
