## [Decision Letter · Decision Letter 0]

16 Oct 2019

PONE-D-19-20938

Musical expertise generalizes to superior temporal scaling in a Morse code tapping task

PLOS ONE

Dear Mr. Slayton,

Thank you for submitting your manuscript to PLOS ONE. After careful consideration, we feel that it has merit but does not fully meet PLOS ONE’s publication criteria as it currently stands. Therefore, we invite you to submit a revised version of the manuscript that addresses the points raised during the review process. In general, both reviewers were positive about the methods and results. One reviewer felt that some additional context in the introduction, addressing the novel contribution of the current study relative to existing studies, as well as the rationale for the task/stimulus, would be helpful. Figure quality was also mentioned by both reviewers. Overall the comments seem addressable to me.

We would appreciate receiving your revised manuscript by Nov 30 2019 11:59PM. To enhance the reproducibility of your results, we recommend that if applicable you deposit your laboratory protocols in protocols.io, where a protocol can be assigned its own identifier (DOI) such that it can be cited independently in the future. For instructions see: http://journals.plos.org/plosone/s/submission-guidelines#loc-laboratory-protocols

We look forward to receiving your revised manuscript.

Kind regards,

Jessica Adrienne Grahn

Academic Editor

PLOS ONE

Journal Requirements:

1. Thank you for including your ethics statement:  

"All experiments were run in accordance with the University of California Human Subjects Guidelines and each subject gave verbal informed consent prior to participation."

Please amend your current ethics statement to confirm that your named institutional review board or ethics committee specifically approved this study.

2.

We note that you have indicated that data from this study are available upon request. PLOS only allows data to be available upon request if there are legal or ethical restrictions on sharing data publicly. For more information on unacceptable data access restrictions, please see http://journals.plos.org/plosone/s/data-availability#loc-unacceptable-data-access-restrictions.

Reviewers' comments:

Reviewer's Responses to Questions

**Comments to the Author**

1. Is the manuscript technically sound, and do the data support the conclusions?

Reviewer #1: Yes

Reviewer #2: Yes

2. Has the statistical analysis been performed appropriately and rigorously? 

Reviewer #1: Yes

Reviewer #2: Yes

3. Have the authors made all data underlying the findings in their manuscript fully available?

Reviewer #1: Yes

Reviewer #2: Yes

4. Is the manuscript presented in an intelligible fashion and written in standard English?

Reviewer #1: Yes

Reviewer #2: Yes

5. Review Comments to the Author

Reviewer #1: The present study investigated effects of motor training of temporal scaling in musicians and non-musicians. The learning curve differed between the two groups. At the end of learning, musicians were better at reproducing the learned Morse code pattern at faster and slower speeds compared with the non-musicians. The study investigated a novel and intriguing issue in the field of motor control. However, I have several concerns particularly on the Introduction and Methods as follows.

First, the present Introduction is currently not clear for readers with respect to significance of the study. This can be largely improved by arguing the previous studies relevant to this study, which seems insufficient in the present manuscript. For example, temporal scaling has been already studied in musicians (e.g. Furuya and Soecthing 2012 J Neurophysiol), and therefore to argue them in the Introduction will make both novelty and significance of the study clearer. Similarly, neurophysiological studies such as bird songs have been also studied this issue extensively (Long and Fee 2008 Nature). Such neurophysiological studies will aid in deepening the understanding of what mechanisms can underlie the temporal scaling of sequential motor actions.

Second, it is better to provide a rationale of investigating the Morse code production task rather than piano playing, even though the latter is more naturalistic to the present participants. In general, a learning task used for sequential movements is a serial reaction time task (e.g. Karni et al. 1998 PNAS). As far as I know, the present motor task is not as frequently used as such a task, which makes me wonder a particular reason behind.

Related to this, the present Discussion can be largely improved by arguing an issue of signal-dependent noise in motor commands (Harris and Wolpert 1998 Nature) and/or Fitt's law. When moving faster, effects of this noise is increased, and vice versa. This neurophysiological mechanism will be crucial to shed light on improved temporal scaling with training.

There are some minor comments.

Abstract: page 8, line 7

What do you mean by "controls"?

Introduction: page 9 second paragraph

While the authors state "It has also been shown that temporal precision can be improved by increasing movement speed (6).", this claim contradicts with the Fitts' Law. Please explain more.

Figures are overall coarse in terms of resolution, which is better to be improved. Also I wonder if the color figures are really necessary.

Reviewer #2: Review of „Musical expertise generalizes to superior temporal scaling in a Morse code tapping task“

by Slayton, Romero-Sosa, Stone, Buonomano & Viskontas

This paper describes a highly original experimental study which contrasts the timing performance of a simple rhythmic pattern (actually the word time in Morse code) by expert musicians vs. non-musicians. Participants practiced the corresponding temporal pattern at a fixed speed in 15 blocks of 15 trials each for four days, with detailed feedback about the accuracy of each production as compared to the target pattern. On day 5, participants were to produce the trained rhythmic pattern at each of three speeds: the original one, twice, or half as fast as the trained pattern. The goal of the study was to test the hypothesis that professional musicians are superior in temporal scaling, i.e., to transfer a rhythmic temporal pattern acquired at one fixed tempo to slower and to faster tempi. The main findings are summarized by the authors as

„Although there was no significant difference in performance at the end of training at the standard speed, musicians were significantly better at temporal scaling—i.e., at reproducing the learned Morse code pattern at faster and slower speeds. (…) Both musicians and non-musicians exhibited a Weber-speed effect, where absolute temporal precision sharpened when producing patterns at the faster speed.“ Not really unexpected..

Minor comments.

Figures. The graphical quality is embarrassingly low. Here is an enlarged snapshot of task and stimuli:

Who can decode the details?

P. 3, 3rd par: „temporal variability“: as assessed by the standard deviation?

p. 4, top: Define NMRSE – not all readers will be familiar with it. Report whether you normalize by mean or by range.

p. 8, 2nd par: replace „time-independent“ by „tempo independent“.

p. 8, 2nd par: „Between each element that comprises a letter and the first element of another letter was a dot length pause, and between the last element of a letter and the first element of another letter was a dash-length pause.“ Why don’t you give just the sequence time-intervals to produce?

For figure see doc-file.

6. PLOS authors have the option to publish the peer review history of their article (what does this mean?). If published, this will include your full peer review and any attached files.

Reviewer #1: No

Reviewer #2: No

---

## [Author Response · Author response to Decision Letter 0]

18 Nov 2019

Our Response to Reviewers document has been attached. We are grateful to the reviewers and editor for their help.

---

## [Decision Letter · Decision Letter 1]

12 Dec 2019

Musical expertise generalizes to superior temporal scaling in a Morse code tapping task

PONE-D-19-20938R1

Dear Dr. Slayton,

We are pleased to inform you that your manuscript has been judged scientifically suitable for publication and will be formally accepted for publication once it complies with all outstanding technical requirements.

With kind regards,

Jessica Adrienne Grahn

Academic Editor

PLOS ONE

Additional Editor Comments (optional):

Reviewers' comments:

Reviewer's Responses to Questions

**Comments to the Author**

1. If the authors have adequately addressed your comments raised in a previous round of review and you feel that this manuscript is now acceptable for publication, you may indicate that here to bypass the “Comments to the Author” section, enter your conflict of interest statement in the “Confidential to Editor” section, and submit your "Accept" recommendation.

Reviewer #1: All comments have been addressed

Reviewer #2: All comments have been addressed

2. Is the manuscript technically sound, and do the data support the conclusions?

Reviewer #1: Yes

Reviewer #2: Yes

3. Has the statistical analysis been performed appropriately and rigorously? 

Reviewer #1: Yes

Reviewer #2: Yes

4. Have the authors made all data underlying the findings in their manuscript fully available?

Reviewer #1: Yes

Reviewer #2: Yes

5. Is the manuscript presented in an intelligible fashion and written in standard English?

Reviewer #1: Yes

Reviewer #2: Yes

6. Review Comments to the Author

Reviewer #1: The authors revised the manuscript in a satisfactory manner, which improved readability a lot. I therefore believe the current manuscript deserves to be published as it is.

Reviewer #2: The authors have taken care of all of my comments and recommendation (most of which were minor anyway.

7. PLOS authors have the option to publish the peer review history of their article (what does this mean?). If published, this will include your full peer review and any attached files.

Reviewer #1: No

Reviewer #2: Yes: Dirk Vorberg

---

## [Editor Report · Acceptance letter]

18 Dec 2019

PONE-D-19-20938R1 

Musical expertise generalizes to superior temporal scaling in a Morse code tapping task 

Dear Dr. Slayton:

I am pleased to inform you that your manuscript has been deemed suitable for publication in PLOS ONE. Congratulations! Your manuscript is now with our production department. 

With kind regards,

on behalf of

Dr Jessica Adrienne Grahn 

Academic Editor

PLOS ONE